# Neutrophil stimulation with citrullinated histone H4 slows down calcium influx and reduces NET formation compared with native histone H4

**Lai Shi**[1,2], **Karen Aymonnier**[1,2], **Denisa D. Wagner**[1,2,3] *

**1** Program in Cellular and Molecular Medicine, Boston Children's Hospital, Boston, Massachusetts, United States of America, **2** Department of Pediatrics, Harvard Medical School, Boston, Massachusetts, United States of America, **3** Division of Hematology/Oncology, Boston Children's Hospital, Boston, Massachusetts, United States of America

* denisa.wagner@childrens.harvard.edu

**Data Availability Statement:** All relevant data are within the manuscript and its Supporting Information files.

**Funding:** This work was supported by the National Heart, Lung, and Blood Institute of the National

## Abstract

Peptidylarginine deiminase 4 (PAD4) catalyzes posttranslational modification of many target proteins through converting protein arginine or mono-methylarginine to citrulline. Neutrophil extracellular trap (NET) formation is the most dramatic manifestation of PAD4-mediated hypercitrullination reaction in neutrophils, which is characterized by the release of nuclear chromatin to form a chromatin network in the extracellular space. Histones H4, one of the major protein components of chromatin, is released into the extracellular space during sepsis, trauma, and ischemia-reperfusion injury and can also be released during the process of NET formation, along with its citrullinated form. The present study showed that histone H4 can induce NET formation in a calcium and PAD4 dependent manner. Histone H4 caused permeabilization of the neutrophil membrane and sustained rise in intracellular calcium that is necessary for activation of PAD4. In comparison, citrullinated histone H4 induced less calcium influx compared with its native form, leading to reduced NET formation. These studies suggest that citrullinated histone H4 could serve as a brake in the pathology of NETs, slowing down the vicious circle between histone H4 and NETs.

## Introduction

Neutrophil extracellular traps (NETs) were initially described as part of the innate immune response to microbes [1] and increasing evidence shows their presence even under sterile conditions [2, 3]. NETs are formed through the release of antimicrobial proteins anchored to a chromatin meshwork [1, 4]. During the past several years, a link between excessive NETs and autoimmunity has been proposed [5, 6]. NETs also form and play a role during fibrosis, ischemic stroke, preeclampsia, thrombosis, cancers, and other inflammatory diseases [2, 3, 7–11].

As a major prerequisite for NET formation, peptidylarginine deiminase 4 (PAD4) catalyzes hypercitrullination reaction on histones and mediate chromatin decondensation, which

Institutes of Health (grant R35 HL135765 to D.D. W.).

**Competing interests:** The authors have declared that no competing interests exist.

promotes the release of NETs [12, 13]. This calcium-dependent epigenetic enzyme causes a citrullination reaction by converting arginine or mono-methylarginine to citrulline on histones [14, 15]. The essential role of PAD4 in NET formation can be revealed by the fact that extracellular trap structures can also be formed by nonhematopoietic cells overexpressing PAD4, such as cancer cells [16, 17].

Many different stimuli can induce NET formation, such as calcium ionophore ionomycin, phorbol myristate acetate (PMA), microbial products like lipopolysaccharide (LPS), crystals, cytokines, and others [18]. Histones, as the major component of NETs, are released into the extracellular space and elevated during sepsis, trauma, and ischemia-reperfusion injury through NET formation or other lytic cell death [19–22]. Although recent reports suggested that extracellular histones are toxic and can also induce NET release [22–24], the involvement of PAD4 in histone toxicity and the effects of citrullination of histones on propagation of NET formation are not yet clear.

Here, we demonstrated that histone H4 induced NET formation in a PAD4 and calcium dependent manner. In comparison, citrullinated histone H4 released during the process of NET formation induced NETs to a lesser extent. Compared with native histone H4, its citrullinated form induced less calcium influx, thus leading to reduced activation of PAD4. Our study suggested that the vicious circle between NETs and histone H4 can be slowed down by citrullination on histone H4, preventing the non-stop, feed-forward loop between histone H4-induced NET formation and increased histone H4 levels from NET release.

## Materials and methods

### Cell culture

HL-60 cells (ATCC, catalog #: CCL-240) were cultured in RPMI 1640 medium containing L-glutamine (Gibco) supplemented with 25 mmol/L HEPES, 1% penicillin/streptomycin and 10% heat-inactivated FBS. Cells were split at $1.5 \times 10^5$ cells/mL. HL-60 cell differentiation was performed for 7 days, by using culture media containing 1.25% DMSO [25].

### Mice

Wild-type (C57BL/6J: stock no. #000664, n = 25) mice were obtained from Jackson Laboratory (Bar Harbor, ME, USA). *Padi4* knockout mice (n = 6) were originally generated by Y. Wang [12] and back-crossed with C57BL/6J mice in the Wagner laboratory. Animals were group-housed, at a maximum of five per cage, on a 12-hour light/dark cycle in the animal facility of Boston Children's Hospital. Food and water were provided ad libitum. All experimental animal procedures in this study were approved by the Institutional Animal Care and Use Committee of Boston Children's Hospital under the protocol numbers 20-01-4096R and 20-02-4097R.

### Human neutrophil isolation

The experimental procedure was approved by the Office of Clinical Investigations at Boston Children's Hospital (protocol number IRB-P00003283). Written informed consent was provided by donors. Blood was drawn from healthy donors in EDTA-coated vacutainers (Becton Dickinson), and blood samples were de-identified prior to isolation. Neutrophils were isolated using gradient centrifugation, as described elsewhere [26]. Cells were resuspended in phenol red free RPMI 1640 medium supplemented with HEPES, assessed for purity by Wright-Giemsa stain (Sigma-Aldrich), and adjusted to the desired cell density.

## Mouse neutrophil isolation

Upon deep isoflurane anesthesia, peripheral mouse neutrophils were collected by drawing blood from the retro-orbital plexus in 1 mL of Ethylenediaminetetraacetic acid (EDTA) anti-coagulant buffer supplemented with 1% endotoxin-free BSA in sterile PBS, before mice were sacrificed by cervical dislocation. Subsequently, peripheral neutrophils were isolated by Percoll (Sigma-Aldrich) gradient centrifugation, as described elsewhere [26]. Neutrophils were then resuspended in phenol red-free RPMI 1640 medium supplemented with 10 mM HEPES, and cell purity was assessed by Wright-Giemsa stain. After the neutrophil count was determined, the required cell density was adjusted by adding HEPES supplemented RPMI 1640 medium.

## NETs assay

For NETs induction, cells were simulated with 4 μmol/L calcium ionophore ionomycin (Invitrogen), 20 ng/μL native (NEB) or citrullinated histone H4 (custom), or 4 ng/μL recombinant human PAD4 (custom) control for 4 h at 37°C and 5% CO2 in phenol red-free RPMI 1640 medium supplemented with 10 mmol/L HEPES. For PAD4 or calcium inhibition, cells were incubated with 200 μmol/L Cl-amidine (Cayman) or 10 μmol/L BAPTA, AM (Invitrogen) for 30 minutes before native histone H4 treatment.

## PAD4 protein production

To generate an N-terminal His6-tagged fusion protein, human PAD4 cDNA was cloned in the vector pNIC28 for bacterial expression, a generous gift from the Structural Genomics Consortium (University of Oxford, UK).

The PAD4 expression construct encoding a His6-PAD4 fusion protein was freshly transformed into E. coli Rosetta cells (Novagen). A single colony was then used to prepare starter cultures, which were then used to inoculate 1 L of autoinduction Terrific Broth media (Formedium). Cells were incubated at 37°C with shaking (300 rpm) for 6 hours and then overnight at 30°C to allow protein expression. The cells were harvested by centrifugation (7000 g for 10 minutes) and frozen overnight at -80°C. The cell pellet was thawed on ice for 15 minutes, at which point 30 mL of lysis buffer (50 mmol/L $Na_2HPO_4$, 300 mmol/L NaCl, 10 mmol/L Imidazol, 0.1% Triton and protease inhibitor cocktail) was added. The cells were then resuspended at 4°C for over 30 minutes. Cell suspensions were sonicated on ice for 3 cycles of a 15 s pulse followed by a 120 s cooling interval. Cell debris was removed by centrifugation at 14000 g for 30 min. The resulting His6-PAD4 fusion protein was purified with Ni-NTA Fast Start Kit (Qiagen) according to the manufacturer's protocol. To remove excess imidazole and to exchange the buffer against 20 mmol/L HEPES, 150 mmol/L NaCl, 20 mmol/L DTT, and 10% glycerol, centrifugal filter units (Amicon Ultra-15, Merk Millipore) were used. The commercial endotoxin affinity EndoTrap HD 1 mL column (LIONEX) was used for endotoxin removal. Protein is then aliquoted and stored at -80°C.

Lastly, concentration is assessed by a BCA assay, the purification is checked by Coomassie blue staining gel, and the activity of PAD4 was checked with a BAEE colorimetric assay [27].

## In vitro citrullination assay

To make citrullinated histone H4, 100 ng/μL native histone H4 was incubated with 20 ng/ μL PAD4 for 3 h at 37°C in citrullination buffer (50 mmol/L Tris-HCl, 4 mmol/L $CaCl_2$, pH 7.6). Citrullinated histone H4 was then concentrated to 500 ng/μL with centrifugal filter units for immunostaining to reduce the buffer effect and can be used directly for Fluo-4 assay below.

When compared with citrullinated histone H4, native histone H4 was also incubated in the same reaction buffer for 3 h and concentrated or not accordingly.

## Immunostaining

Cells after stimulation were fixed for 15 min at room temperature with 4% paraformaldehyde (Electron Microscopy Sciences) in Phosphate Buffer Saline (PBS) with 0.1% Triton X-100 (PBST). Cells were washed once for 5 min before being blocked for 30 min in blocking solution (2% BSA in PBST). Cells were stained overnight at 4°C with primary antibodies diluted in blocking solution, then washed before being incubated with Alexa Fluor 488 (1:1000, Invitrogen, catalog #: A-11008, lot #: 2140660) and DAPI (1:1000, Sigma) diluted in PBST for 2 h at room temperature. Cells were washed with PBST before being mounted in mounting media (Electron Microscopy Sciences). The following primary antibodies were used: anti-H3Cit (1:1000, abcam, catalog #: ab5103, lot #: GR262518-1) and anti-H4Cit3 (1:200, EMD Millipore, catalog #: 07–596, lot #: 3321101). Images were collected using a Zeiss Axiovert 200M Fluorescence/Live cell Imaging Microscope and processed with Adobe Photoshop and Illustrator.

## Western blot

For western blot analysis, cells were lysed in RIPA buffer (Thermo Fisher Scientific) supplemented with protease inhibitor and sonicated for 15 minutes in an ice bath. The protein concentration was measured using BCA reagent. The protein samples were then denatured in LDS buffer (Thermo Fisher Scientific) and reducing agent (Thermo Fisher Scientific) for 5 minutes at 95°C. Lysates were separated by 4–12% Bis-Tris gradient gels (Thermo Fisher Scientific), and proteins were electrotransferred on a PVDF membrane using the iBlot system (Thermo Fisher Scientific). Membranes were blocked with 5% BSA in TBST buffer (0.05% Tween-20 in TBS) for 30 min at room temperature and incubated overnight at 4°C with primary antibodies. After incubation with primary antibodies, membranes were washed 3 times with TBST buffer before incubation with HRP-conjugated secondary antibodies for 2 h at 4°C. Then the membranes were further washed 3 times with TBST and subsequently probed with enhanced chemiluminescence (ECL) detection solution (Thermo Fisher Scientific). The following antibodies were used: anti-histone H4 (1:1000, EMD Millipore, catalog #: 07–108, lot #: 2982295), anti-GAPDH (3 μg/mL, Thermo Fisher Scientific, catalog # MA5-15738), anti-PAD4 (1:2000, abcam, catalog #: ab128086, lot #: GR3221322-4), anti-H3Cit (1:1000, abcam, catalog #: ab5103, lot #: GR262518-1), anti-H4Cit3 (1:1000, EMD Millipore, catalog #: 07–596, lot #: 3321101), anti-rabbit HRP conjugated (1:10000, Sigma-Aldrich, catalog #: 12–348, lot #: R138481), and anti-mouse HRP conjugated (1:3000, Bio-Rad, catalog #: #1706516, lot #: 170–6516).

## Measurement of calcium influx

Differentiated HL60 cells or human neutrophils were concentrated to $5 \times 10^6$/mL with phenol red-free RPMI 1640 medium supplemented with 10 mM HEPES and loaded with 5 μg/mL calcium-sensitive dye Fluo-4, AM (Invitrogen) for 15 min at room temperature. Cells were then washed to remove extracellular Fluo-4, AM and resuspended with phenol red-free RPMI 1640 medium supplemented with 10 mM HEPES at $5 \times 10^6$/mL in 25 μL per well. Baseline fluorescent signal was recorded kinetically with filter pair excitation 485/emission 538 for 5 min with an interval of 3 s in 96-well plate using Fluoroskan Microplate Fluorometer (Thermo Scientific). 25 μL cells were then mixed with control, native histone H4, citrullinated histone H4, ionomycin, or ionomycin plus PAD4 protein in 75 μL citrullination buffer. The final concentrations were 75 ng/μL for histones, 4 μmol/L for ionomycin, and 15 ng/μL for PAD4 protein.

The change of fluorescent signal was immediately recorded kinetically for 2 h with an interval of 7.2 s.

## Statistics

For all numerical analyses, results are expressed as mean ± SD. Group comparisons were performed using Student $t$ test. Values of $P < 0.05$ were considered statistically significant. These analyses were carried out using the GraphPad Prism 5 Software (GraphPad Software).

## Results

### Citrullinated histone H4 was released into extracellular space along with its native form in the process of NET formation

The most immediate source of extracellular histones is necrotic cell death, when intracellular content is released due to the rupture of the plasma membrane or apoptotic cells through membrane blebs and nucleosomes [28, 29]. However, neutrophils, by forming NETs, also release histones, among which is the highly toxic histone H4 [1, 30]. The potential differences in the biological function of histone H4 released during different cell death processes are that some histone H4 were citrullinated by PAD4 in the process of NET formation (Fig 1A–1D). Citrullinated histone H4 was released into the extracellular space along with its native form when differentiated HL60 cells (dHL60) or human neutrophils were stimulated with calcium ionophore ionomycin for NET formation (Fig 1A–1D).

### Histone H4–induced NET formation is dependent on PAD4 and calcium

In accordance with previous reports [23, 24], we found that histone H4 was sufficient to induce NET formation in all the three models tested, including dHL60 cells, human peripheral neutrophils, and mouse peripheral neutrophils (Fig 2A, 2C and 2E). It is also comparable to other physiological stimuli such as LPS (~12% for histone H4 vs ~9.5% for LPS [12]). To address whether the observed NET formation induced by histone H4 was dependent on PAD4 activity, we first pretreated cells with pan-PAD inhibitor Cl-amidine (Cla), which inhibits both PAD4 and PAD2 [31], before histone H4 stimulation. Inhibition of PAD activity significantly decreased the incidence of NET formation induced by histone H4 in dHL60 cells and primary human peripheral neutrophils (Fig 2A–2D). The reduced citrullination level within neutrophils after Cla treatment was also verified by western blot using anti-H3Cit and anti-H4Cit3 antibodies (S1 Fig). To test the function of PAD4 specifically, we compared NET formation after H4 stimulation between wild type (WT) and *Padi4* knockout (KO) mouse peripheral neutrophils (Fig 2G and 2H). In comparison to the extensive NET formation in WT mouse neutrophils, citrullination and the formation of NETs were not detected in cells lacking *Padi4* and stimulated by histone H4 (Fig 2G and 2H). Together, these aforementioned analyses indicate that PAD4 activity is required for histone hypercitrullination and NET formation induced by histone H4.

　　PAD4 is a calcium-dependent enzyme, therefore, we further tested whether calcium elevation is required for NET formation under the condition of histone H4 stimulation. To test this idea, we employed an intracellular calcium chelator 1,2-Bis(2-aminopheoxy)ethane-N,N,N', N'-tetraacetic acid tetrakis (acetoxymethyl ester), better known as BAPTA, AM [32], at nontoxic concentrations. After pretreatment with BAPTA, AM, a decrease in the extent of cells undergoing chromatin decondensation was detected (Fig 2A–2F). Moreover, western blot analyses detected reduced levels of H3Cit after treatment with calcium chelator (S1 Fig), suggesting that calcium is important for histone citrullination after histone H4 treatment. These

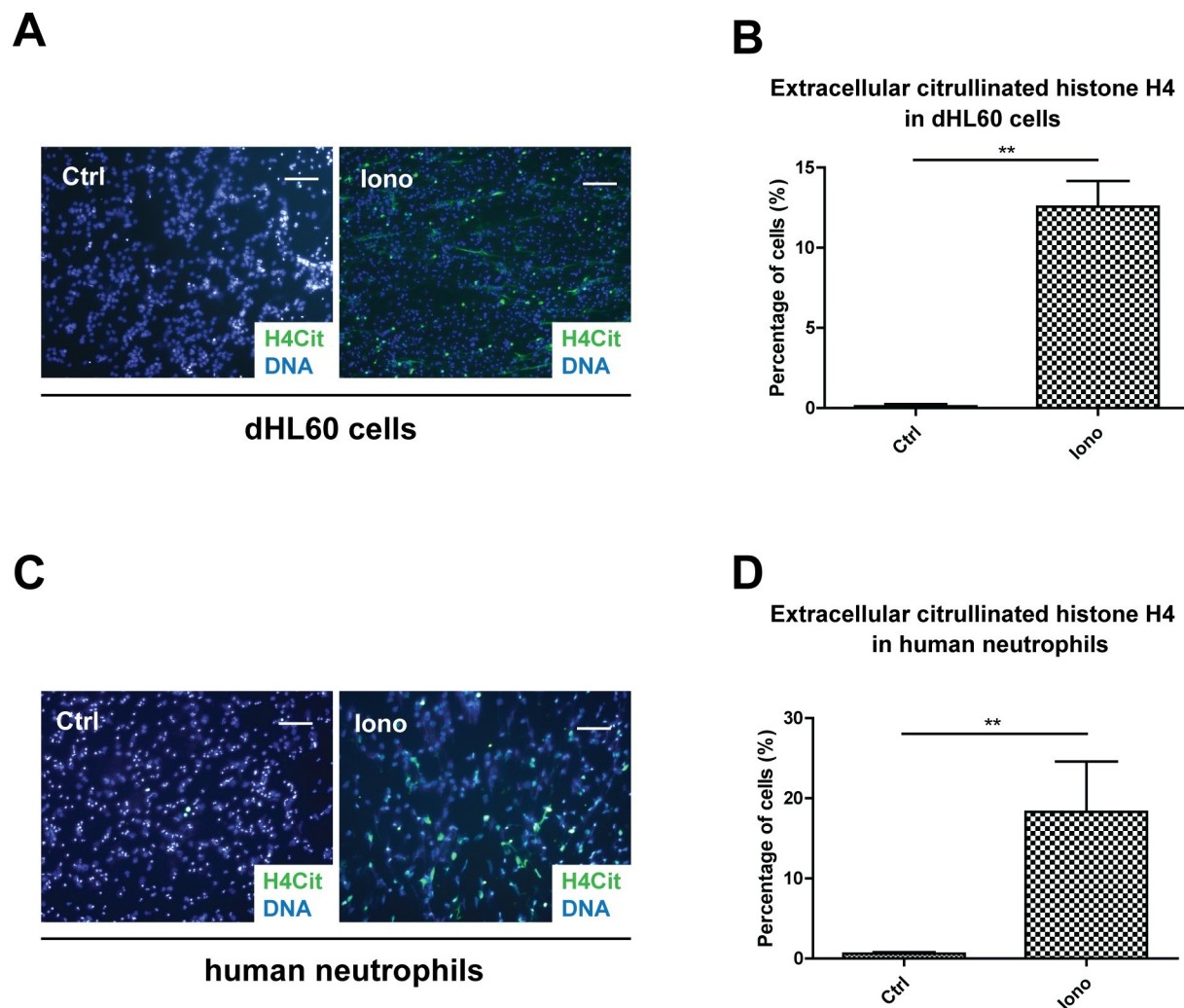

**Fig 1. Citrullinated histone H4 is released into extracellular space in the process of NET formation.** (A) Representative immunofluorescence images of dHL60 cells showing extracellular citrullinated histone H4 staining after 4 hours of ionomycin (iono) treatment with staining for H4Cit (green) and nuclei (blue). Scale bars, 100 μm. (B) Quantification of extracellular histone H4 staining in dHL60 cells after ionomycin treatment. Mean ± SD is shown ($n$ = 3 independent experiments). (C) Representative immunofluorescence images of human neutrophils showing extracellular citrullinated histone H4 staining after 4 hours of ionomycin treatment with staining for H4Cit (green) and nuclei (blue). Scale bars, 100 μm. (D) Quantification of extracellular histone H4 staining in human neutrophils after ionomycin treatment. Mean ± SD is shown ($n$ = 3 independent experiments). **, $P < 0.01$. Ctrl, unstimulated control.

results reveal that neutrophils must achieve sufficient intracellular calcium concentrations to induce PAD4-mediated histone citrullination and chromatin decondensation following histone H4 stimulation.

## Citrullinated histone H4 induced less NET formation compared with native histone H4

Due to the fact that citrullinated histone H4 was also released into extracellular space along with native histone H4 in the process of NET formation (Fig 1, [1]), and native histone H4 is known to induce NET formation (Fig 2, [23, 24]), we next tested the ability of citrullinated histone H4 in induction of NET formation. Native histone H4 was incubated with human PAD4 prepared in our laboratory to citrullinate histone H4 (S2 Fig). Although the citrullinated

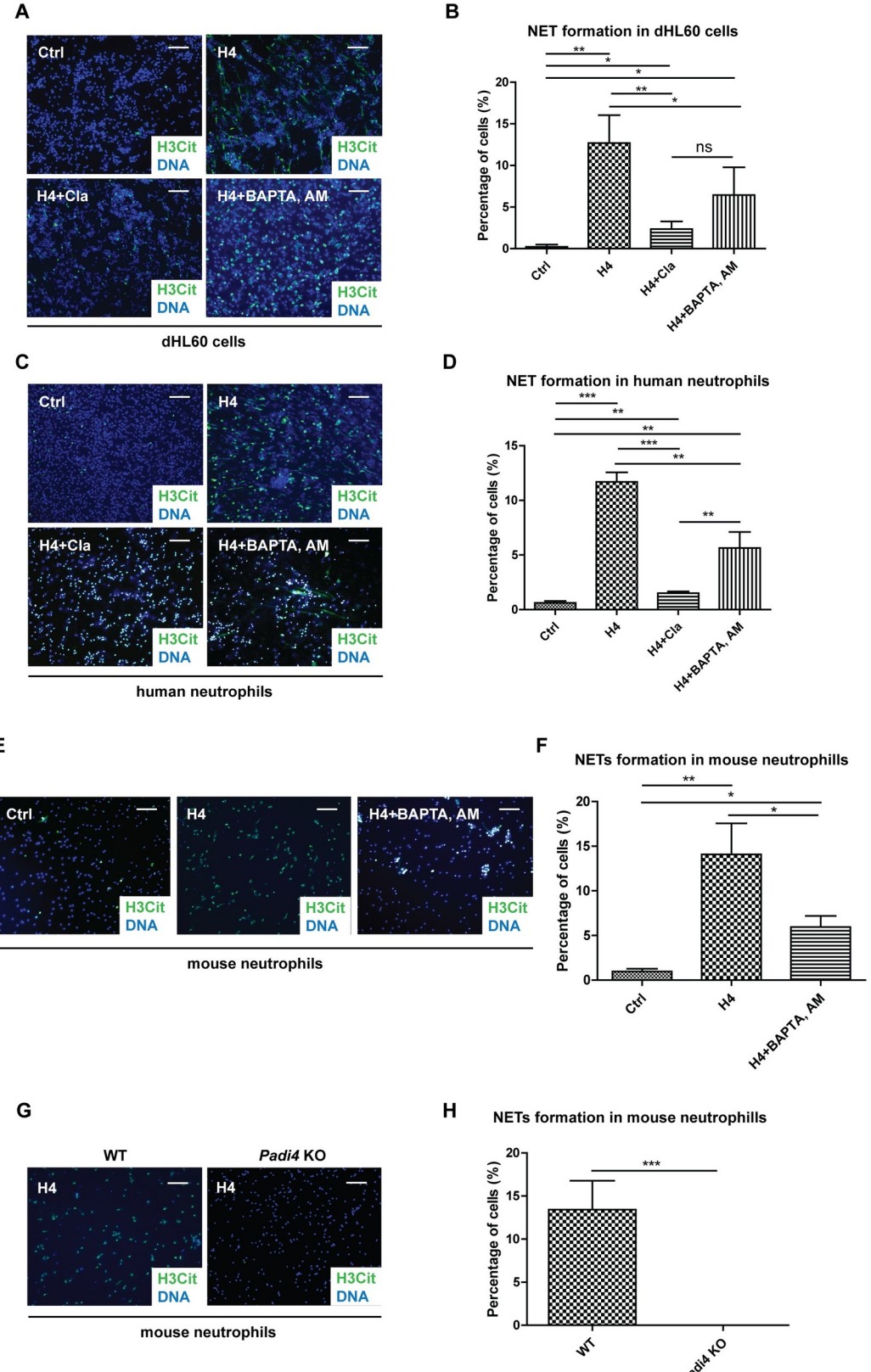

**Fig 2. Histone H4–induced NET formation is dependent on PAD4 and calcium.** (A) Immunofluorescence images revealed reduced histone H4-induced NET formation with pretreatment by PAD inhibitor Cl-amidine or intracellular calcium chelator BAPTA, AM in dHL60 cells. H3Cit (green) and nuclei (blue). Scale bars, 100 μm. (B) Quantification of NET formation in dHL60 cells after the different stimuli in (A). Mean ± SD is shown (*n* = 3 independent experiments). (C) Immunofluorescence images revealed reduced histone H4-induced NET formation with pretreatment by PAD inhibitor Cl-amidine or intracellular calcium chelator BAPTA, AM in human neutrophils. H3Cit (green) and nuclei (blue). Scale bars, 100 μm. (D) Quantification of NET formation in human neutrophils after the different stimuli in (C). Mean ± SD is shown (*n* = 3 independent experiments). (E) Immunofluorescence images revealed reduced histone H4-induced NET formation with pretreatment of intracellular calcium chelator BAPTA, AM in mouse neutrophils. H3Cit (green) and nuclei (blue). Scale bars, 100 μm. (F) Quantification of histone H4-induced NET formation in mouse neutrophils with or without pretreatment of calcium chelator BAPTA, AM. Mean ± SD is shown (*n* = 3 or 4 mice). (G) Immunofluorescence images of wild type and *Padi4* knockout mouse neutrophils after histone H4 treatment. H3Cit (green) and nuclei (blue). Scale bars, 100 μm. (H) Quantification of histone H4-induced NET formation in wild type and *Padi4* knockout mouse neutrophils. Mean ± SD is shown (*n* = 5 for wild type mice and 3 for *Padi4* knockout mice). *, $P < 0.05$; **, $P < 0.01$; ***, $P < 0.001$; ns, not significant. Ctrl, unstimulated control.

histone H4 could still induce NET formation, it was significantly less effective compared with its native form in all three models tested (Fig 3). To avoid the confounding effect of the PAD4 protein in the citrullination reaction, we tested that PAD4 protein alone did not induce NET formation, confirming the observed difference in NET formation is due to citrullination of histone H4 (Fig 3A–3C). In addition, neither native nor citrullinated histone H4 influenced PAD4 expression levels in neutrophils as observed by western blot (Fig 3C). In contrast, native histone H4 induced apparent PAD4 activation revealed by increased citrullination of histone H4 and H3 and the increase in citrullination was reduced in neutrophils stimulated by citrullinated histone H4 (Fig 3C and 3F). In summary, the above experiments suggest that native histone H4 activated PAD4, instead of increasing its expression level, to induce NET formation in neutrophils, and citrullinated histone H4 led to reduced NET formation with lower PAD4 activation.

## Citrullinated histone H4 slows down calcium influx compared with native histone H4

The interaction of histones with different cell types results in an influx of calcium without mobilization of internal calcium stores [33]. Our previous report demonstrated that histones bound to platelets, induced calcium influx, and recruited plasma adhesion proteins, such as fibrinogen, to induce platelet aggregation [30]. Recently, histone H4 was also shown to be able to directly stimulate neutrophil activation through membrane permeabilization and sustained intracellular calcium elevation [34]. The discoveries above and our result with BAPTA, AM, inhibiting histone H4-induced NET formation, prompted us to test whether citrullinated histone H4 has a diminished ability to induce calcium influx in neutrophils.

We loaded neutrophils with a calcium-sensitive fluorescent dye Fluo-4, AM and incubated them with native or citrullinated histone H4. We incubated cells with ionomycin as a positive treatment. Elevated fluorescence after the addition of histone H4 indicated histone H4-induced calcium influx into dHL60 cells (Fig 4A and 4B) and human neutrophils (Fig 4D and 4E). However, citrullinated histone H4 slowed down the influx of calcium compared with native histone H4 (Fig 4A and 4D). The quantification in Fig 4B and 4E showed the cumulative effects by comparing the area under the curves. In addition, the time to reach half of the maximal fluorescent signal was also significantly delayed in citrullinated histone H4 treated dHL60 cells (Fig 4C) and human neutrophils (Fig 4F) compared with its native form. Presence of PAD4 alone in incubation buffer did not change the kinetics of ionomycin-induced calcium influx (S3 Fig), suggesting that citrullination on histone H4 was the key player. In summary, the aforementioned assays along with BAPTA, AM's inhibitory effect on NET formation

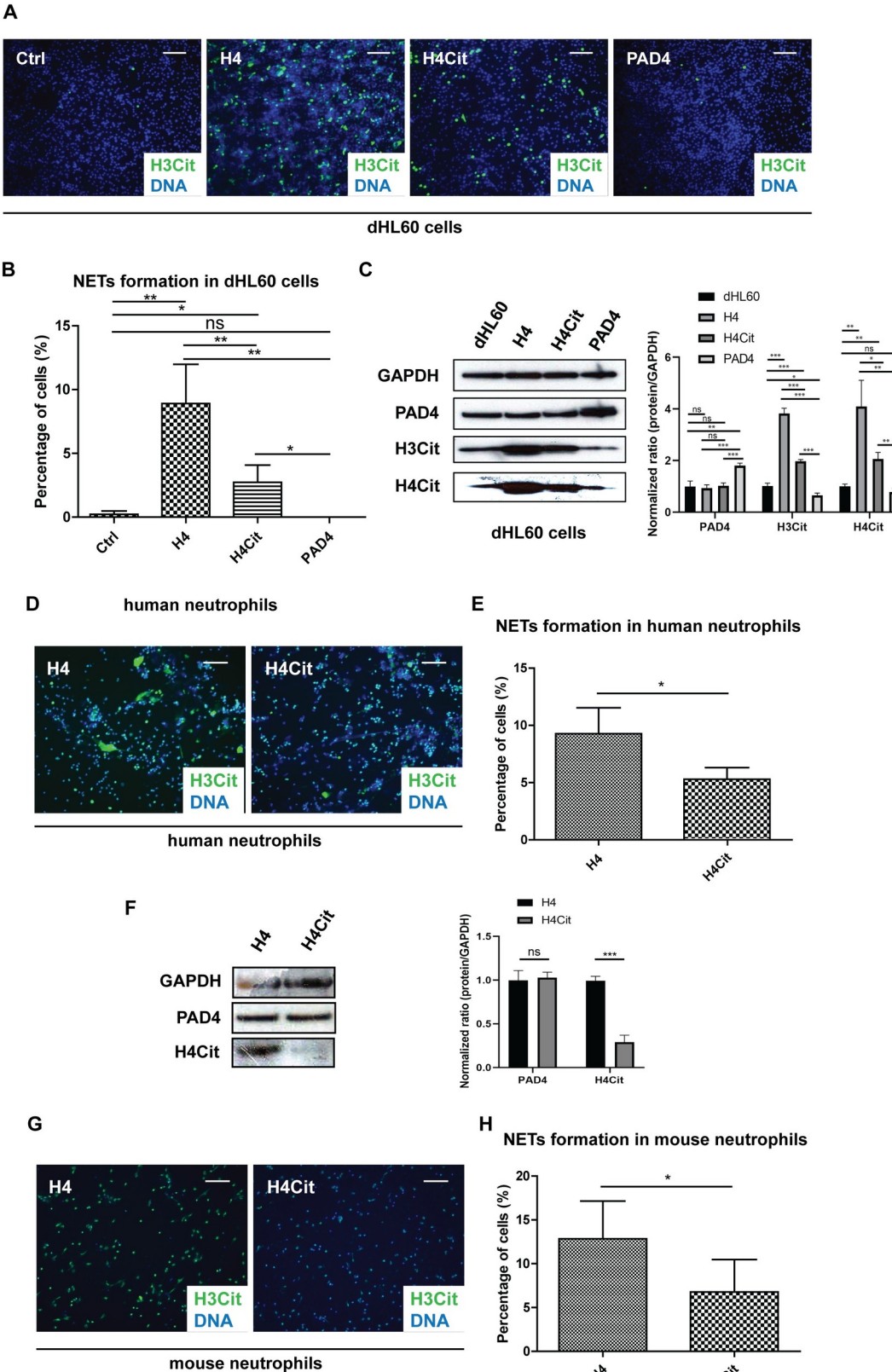

**Fig 3. Citrullinated histone H4 induced less NET formation than native histone H4.** (A) Immunofluorescence images of NET formation in dHL60 cells after treatments with native or citrullinated histone H4 or PAD4 alone as control. H3Cit (green) and nuclei (blue). Scale bars, 100 μm. (B) Quantification of NET formation in dHL60 cells after treatments with native or citrullinated histone H4 or PAD4 alone as control. Mean ± SD is shown (*n* = 3 for controls and 5 for histone H4 treatments). (C) PAD4, H3Cit, and H4Cit protein levels were determined by western blot analysis in dHL60 cells after treatments with native or citrullinated histone H4 or PAD4 alone as control. The results are representative of three experiments. (D) Immunofluorescence images of NET formation in human neutrophils after treatment with native or citrullinated histone H4. H3Cit (green) and nuclei (blue). Scale bars, 100 μm. (E) Quantification of NET formation in human neutrophils after treatment with native or citrullinated histone H4. Mean ± SD is shown (*n* = 4 independent experiments). (F) PAD4 and H3Cit protein levels were determined by western blot analysis in human neutrophils after treatment with native or citrullinated histone H4. The results are representative of three experiments. (G) Immunofluorescence images of NET formation in mouse neutrophils after treatment with native or citrullinated histone H4. H3Cit (green) and nuclei (blue). Scale bars, 100 μm. (H) Quantification of NET formation in mouse neutrophils after treatment with native or citrullinated histone H4. Mean ± SD is shown (*n* = 5 different fields from 3 independent experiments). *, $P < 0.05$; **, $P < 0.01$; ns, not significant. Ctrl, unstimulated control.

suggested that citrullinated histone H4 is a weaker calcium influx inducer compared with native histone H4 and, thus, it led to reduced NET formation.

## Discussion

Histones are integral structural components of chromatin and, after injury or in disease can also be released into extracellular space. This occurs through NET formation, along with other origins, such as dying or activated cells during sepsis, trauma, acute respiratory distress syndromes (ARDS), cancer and other conditions [20, 22, 35–38]. Despite their structural functionality in forming nucleosome complexes, these essential nuclear proteins turn into damage-associated molecular patterns (DAMPS) with potent cytotoxic effects when released extracellularly. For example, a study authored by Xu *et al.* showed that extracellular histones, mainly H3 and H4 subunits, may cause endothelial cell death [22]. Histones induce the release of von Willebrand factor (VWF) from Weibel-Palade bodies (WPBs) in endothelial cells [39] and cause platelet activation and thrombocytopenia [30]. Systemic histone release was linked to microvascular thrombosis, as microthrombi have been observed in lung and kidney sections of histone-treated mice [36, 40]. Epigenetic post-translational modification of nuclear histones is a classical mechanism of histone-mediated control of cell genome. However, histone modification by citrullination has not been examined in terms of their toxicity and ability to induce NET formation.

Our finding that citrullination of histone H4 delayed calcium influx into neutrophils compared with its native form could be in the future extended to studies of other cell types such as endothelial cells and platelets. Due to the pivotal role of calcium signaling, the same comparisons on, for example, platelet aggregation and endothelial cell activation are worth investigating. How histone H4 binds to neutrophils is not clear. On the one hand, histone H4 can bind directly to the plasma membrane of cells by interactions with lipids [41, 42], and histones bound to liposomes were reported to form pores [41]. On the other hand, histone binding to cells may open existing channels [33]. In our study, we did not analyze whether the binding to membrane is weaker when histone H4 is citrullinated. The citrullination modification on positively charged histone arginine could alter the structure and reduced the positive charge of histones, and thus interfere in their interaction with or insertion into the plasma membrane.

The limitation of our study is that we did not determine what was the percentage of histone H4 molecules that were modified by PAD4 during citrullination *in vitro* and which were the critical arginines implicated. While likely not all molecules were citrullinated, the impact of the modifications on reducing histone H4 toxicity was significant. Similarly, it will be important to determine the fraction of histones that are citrullinated during NETosis and whether

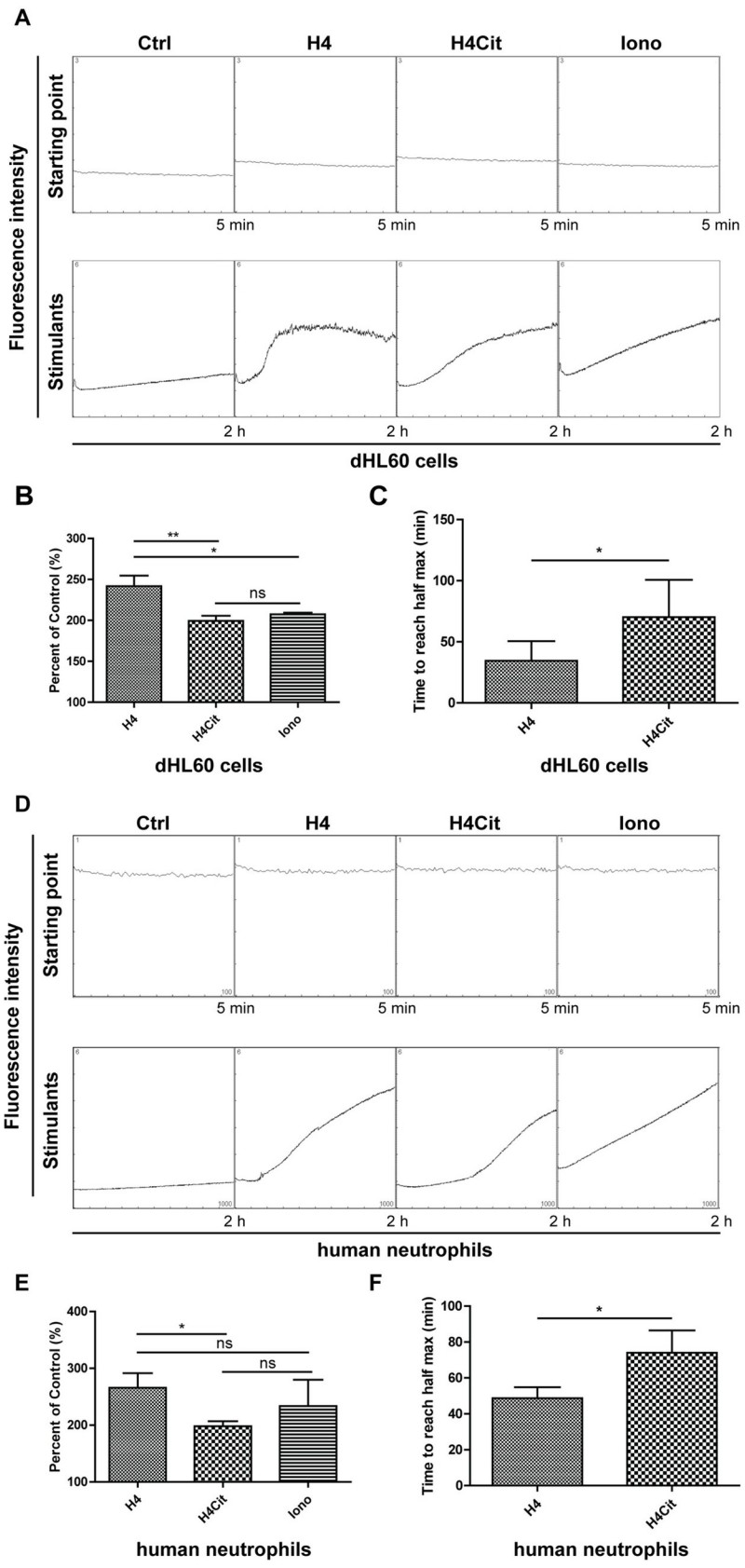

**Fig 4. Citrullinated histone H4 induced less calcium influx than native histone H4.** (A) Fluo-4 kinetic fluorescent readings of dHL60 cells in resting condition (Starting point) and during 2 h of above indicated treatments (Stimulants). (B) Quantification of area under the curve in (A). The area under the curve from histone H4 or ionomycin treatment was normalized to control treatments. Mean ± SD is shown ($n = 4$ independent experiments). (C) Quantification of time to reach half of the maximal signal from native and citrullinated histone H4 treatments in (A). Mean ± SD is shown ($n = 4$ independent experiments). (D) Fluo-4 kinetic fluorescent readings of human neutrophils in resting condition (Starting point) and during 2 h of above indicated treatments (Stimulants). (E) Quantification of area under the curve in (D). The area under the curve from histone H4 or ionomycin treatment was normalized to control treatments. Mean ± SD is shown ($n = 5$ independent experiments). (F) Quantification of time to reach half of the maximal signal from native and citrullinated histone H4 treatments in (D). Mean ± SD is shown ($n = 5$ independent experiments); *, $P < 0.05$, **, $P < 0.01$; ns, not significant. Ctrl, unstimulated control.

this fraction and the identity of histones modified vary with stimulants inducing NET formation.

Although PAD4 may not be required for all types of NET formation [43], histone H4-induced NET formation does depend on the activity of PAD4. Histone H4 permeabilizes the cell membrane, opening the "doors" for calcium entrance. Then, the elevation of intracellular calcium activates PAD4, which is essential for H4-mediated NET formation. Our results show that PAD4 may play a dual role in histone H4-mediated NETosis. It is necessary to produce NETs at early times in injury when free H4 is generated. However the citrullination of H4 during NETosis then dampens its pro-NET activity thus preventing excessive NET formation.

## Conclusions

In this study, we showed that histone H4 induced NET formation in a PAD4 and calcium dependent manner. Citrullinated histone H4 stimulated less calcium influx than its native form, thus leading to reduced NET formation. Our results suggested that citrullination of histone H4 serves as a brake to slow down the vicious loop between histone H4-induced NETs and histone H4 release in the process of NET formation.

## Supporting information

**S1 Fig. Histone H4-induced NET formation is dependent on PAD enzyme and calcium.** H3Cit and H4Cit protein levels were determined by western blot analysis in dHL60 cells after treatments with histone H4 or Cl-amidine + histone H4. The results are representative of three experiments. H3Cit protein level was determined by western blot analysis in dHL60 cells after treatments with histone H4 or BAPTA, AM + histone H4. The results are representative of three experiments. *, $P < 0.05$; ***, $P < 0.001$; ns, not significant. Ctrl, unstimulated control dHL60 cells.
(TIF)

**S2 Fig. Verification of citrullinated histone H4 protein.** Citrullination of histone H4 with PAD4 was verified by western blot analysis. The results are representative of seven experiments. ***, $P < 0.001$.
(TIF)

**S3 Fig. PAD4 alone does not influence calcium influx induced by ionomycin.** Fluo-4 kinetic fluorescent readings of dHL60 cells in resting condition (Starting point) and during 2 h of indicated treatments (Stimulants). The area under the curve from ionomycin or ionomycin + PAD4 treatments were normalized to control treatments for comparison. Mean ± SD is shown ($n = 3$ independent experiments); ns, not significant.
(TIF)

**S1 Raw images.**
(TIF)

## Acknowledgments

The authors acknowledge the meticulous help in preparation of the manuscript by Tiffany Frary. We thank Dr. Patrick Münzer and Dr. Shoichi Fukui for support in isolating human neutrophils and Long Chu and Sarah Gutch for mouse husbandry.

## Author Contributions

**Conceptualization:** Lai Shi, Denisa D. Wagner.

**Data curation:** Lai Shi.

**Formal analysis:** Lai Shi.

**Funding acquisition:** Denisa D. Wagner.

**Investigation:** Lai Shi, Karen Aymonnier, Denisa D. Wagner.

**Methodology:** Karen Aymonnier, Denisa D. Wagner.

**Resources:** Karen Aymonnier.

**Supervision:** Denisa D. Wagner.

**Validation:** Denisa D. Wagner.

**Writing – original draft:** Lai Shi.

**Writing – review & editing:** Karen Aymonnier, Denisa D. Wagner.

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
