## [Decision Letter · Decision Letter 0]

10 Mar 2021

PONE-D-21-02995

Neutrophil stimulation with citrullinated histone H4 slows down calcium influx and reduces NET formation compared with native histone H4

PLOS ONE

Dear Dr. Wagner,

Thank you for submitting your manuscript to PLOS ONE. It interested both the Reviewers and Editor. After careful consideration, we feel that it has merit but does not fully meet PLOS ONE’s publication criteria as it currently stands. Therefore, we invite you to submit a revised version of the manuscript that addresses the points raised during the review process.

In particular, enlarged fluorescence images are necessary, as well as a consolidated statistic analysis.

We look forward to receiving your revised manuscript.

Kind regards,

Michel Simon, Ph. D.

Academic Editor

PLOS ONE

Additional Editor Comments:

the Authors conclusively demonstrated that the citrullinated form of H4 is less effective than the native form to induce NETosis in HL60 cells and human primary neutrophils, because of a lower induced calcium influx. Some of the data concerning mouse neutrophils (Fig 2E and 2G) are less convincing (extra-cellular CitH3 is hardly visible), and better images should be shown. Please, could the Authors also quantify immunodetection data shown on Fig S1A.

Journal Requirements:

2. We noticed you have some minor occurrence of overlapping text with the following previous publication, which needs to be addressed:

- https://pubmed.ncbi.nlm.nih.gov/32193354/

The text that needs to be addressed involves the first few sentences of the abstract.

In your revision ensure you cite all your sources (including your own works), and quote or rephrase any duplicated text outside the methods section.

Further consideration is dependent on these concerns being addressed.

3. Please ensure you have discussed any potential limitations of your study in the Discussion.

4. Please provide the product number and any lot numbers of the antibodies purchased for your study.

5. At this time, we request that you  please report additional details in your Methods section regarding animal care, as per our editorial guidelines:

(a) Please state the number of mice used in the study

(b) Please provide details of animal welfare (e.g., shelter, food, water, environmental enrichment)

(c) Please include the method of euthanasia; please specify whether mice were recovered following retro-orbital bleeding.

Thank you for your attention to these requests.

6. In your Methods section, please provide additional details regarding the cell lines used in your study and ensure you have described the source.

For more information regarding PLOS' policy on materials sharing and reporting, see https://journals.plos.org/plosone/s/materials-and-software-sharing#loc-sharing-materials, and for more information on PLOS ONE's guidelines for research using cell lines, see https://journals.plos.org/plosone/s/submission-guidelines#loc-cell-lines

7. Please provide additional details regarding participant consent.

In the ethics statement in the Methods and online submission information, please ensure that you have specified what type you obtained (for instance, written or verbal, and if verbal, how it was documented and witnessed).

If your study included minors, state whether you obtained consent from parents or guardians.

If the need for consent was waived by the ethics committee, please include this information.

8. PLOS ONE now requires that authors provide the original uncropped and unadjusted images underlying all blot or gel results reported in a submission’s figures or Supporting Information files. This policy and the journal’s other requirements for blot/gel reporting and figure preparation are described in detail at https://journals.plos.org/plosone/s/figures#loc-blot-and-gel-reporting-requirements and https://journals.plos.org/plosone/s/figures#loc-preparing-figures-from-image-files. When you submit your revised manuscript, please ensure that your figures adhere fully to these guidelines and provide the original underlying images for all blot or gel data reported in your submission. See the following link for instructions on providing the original image data: https://journals.plos.org/plosone/s/figures#loc-original-images-for-blots-and-gels.

9. PLOS requires an ORCID iD for the corresponding author in Editorial Manager on papers submitted after December 6th, 2016. Please ensure that you have an ORCID iD and that it is validated in Editorial Manager. To do this, go to ‘Update my Information’ (in the upper left-hand corner of the main menu), and click on the Fetch/Validate link next to the ORCID field. This will take you to the ORCID site and allow you to create a new iD or authenticate a pre-existing iD in Editorial Manager. Please see the following video for instructions on linking an ORCID iD to your Editorial Manager account: https://www.youtube.com/watch?v=_xcclfuvtxQ

10. Please include captions for your Supporting Information files at the end of your manuscript, and update any in-text citations to match accordingly. Please see our Supporting Information guidelines for more information: http://journals.plos.org/plosone/s/supporting-information

Reviewers' comments:

Reviewer's Responses to Questions

**Comments to the Author**

1. Is the manuscript technically sound, and do the data support the conclusions?

Reviewer #1: Yes

Reviewer #2: Partly

2. Has the statistical analysis been performed appropriately and rigorously? 

Reviewer #1: Yes

Reviewer #2: No

3. Have the authors made all data underlying the findings in their manuscript fully available?

Reviewer #1: Yes

Reviewer #2: Yes

4. Is the manuscript presented in an intelligible fashion and written in standard English?

Reviewer #1: Yes

Reviewer #2: Yes

5. Review Comments to the Author

Reviewer #1: The paper analyzes the ability of “native” vs citrullinated histones to induce NETosis in mouse and human neutrophils and in HL60 cells. The results indicate that histone H4 induces NETosis mediating calcium influx and PADi4 activation, while citrullinated H4 has a lower activity, suggesting that citrullinated histones in NET may act as negative feedback mechanism in NET production.

On the whole, the data presented in the paper are of great interest and relevance; only a minor point can be raised.

Under a quantitative point of view, is NET induction by native histones comparable to what obtained with other “physiological” stimuli (for example lps)?

Reviewer #2: The manuscript submitted by Shi et al. addresses the role of H4 citrullination in NET formation. The topic is interesting and the results suggest that the release of citrullinated H4 may constitute a physiological mechanism to slow down the process of NET formation. The manuscript is well written, but it would need additional controls and changes before considering its publication in PlosOne.

Specific comments:

1. In line 192 the authors state that citrullinated H1 is released into the extracellular space along native H4. The authors would need to demonstrate this statement showing an experiment where native H4 is shown to be released into the extracellular space. Moreover, it would be really helpful to determine the proportion H4 modified by citrullination. These results should be added to Figure 1, or as an additional figure.

2. The statistical analysis shown in the figures must be revised to include all possible pairs. For instance, Figure 2B does not show if the percentage of NET formation in the presence of Cla is significant when compared to the control cells. The same problem can be observed in all the figures in panels with more than two bars.

3. The extent of in vitro citrullination must be determined to ensure that all H4 has been modified in the reaction conditions. These results must be included in Figure S2.

4. Regarding the manuscript format, higher resolution images are needed, especially for immunofluorescence and the supplementary file must be revised because at the moment previous changes are marked in red within the document.

6. PLOS authors have the option to publish the peer review history of their article (what does this mean?). If published, this will include your full peer review and any attached files.

Reviewer #1: **Yes: **Paola Migliorini

Reviewer #2: No

---

## [Author Response · Author response to Decision Letter 0]

23 Apr 2021

Dear Dr. Simon,

My colleagues and I appreciate your consideration, as well as the reviewers’ comments, on our manuscript. We have carefully reviewed the comments and made corresponding changes in the manuscript. Our responses are given in a point-by-point manner below. We hope the revised version will now be suitable for publication and look forward to hearing from you.

Thank you very much for your time and help.

Sincerely,

Denisa D. Wagner, Ph.D.

Edwin Cohn Professor of Pediatrics

Program in Cellular and Molecular Medicine

Division of Hematology/Oncology

Boston Children’s Hospital

Harvard Medical School

Responses to Editors and reviewers:

Additional Editor Comments:

the Authors conclusively demonstrated that the citrullinated form of H4 is less effective than the native form to induce NETosis in HL60 cells and human primary neutrophils, because of a lower induced calcium influx. Some of the data concerning mouse neutrophils (Fig 2E and 2G) are less convincing (extra-cellular CitH3 is hardly visible), and better images should be shown. Please, could the Authors also quantify immunodetection data shown on Fig S1A.

Mouse NETs are less extensive compared to human NETs. In our revision, Fig 2E and 2G were replaced with images of different contrast to show the H3Cit staining more clearly. Thank you for suggesting the improvement.

We have now added statistical analysis for all western blot figures in our manuscript, and we also reorganized our figures accordingly to include the new histograms.

Journal Requirements:

Thank you for the reminder to check the revised reference list. We did not cite any retracted articles, and the reference list is complete and correct.

We have now moved the figure captions to locations directly after the paragraph in which they were first cited. We reformatted Supporting Information Citations as “S1 Fig” etc. We also moved Supporting Information to the end of the manuscript. In addition, separate parts of supplementary figures (e.g., S1A and S1B Fig) have also been combined into one file.

2. We noticed you have some minor occurrence of overlapping text with the following previous publication, which needs to be addressed:

- https://pubmed.ncbi.nlm.nih.gov/32193354/

The text that needs to be addressed involves the first few sentences of the abstract.

In your revision ensure you cite all your sources (including your own works), and quote or rephrase any duplicated text outside the methods section.

Further consideration is dependent on these concerns being addressed.

We apologize that the first author adapted a few introductory sentences from his own publication when drafting the initial manuscript. We have now rephrased the sentences of the abstract as highlighted on page 2.

3. Please ensure you have discussed any potential limitations of your study in the Discussion.

We have added more limitations of our study in the Discussion. We did not analyze how histone H4 binds to neutrophils and whether the binding of citrullinated histone H4 to membrane is weaker, now highlighted on page 17. We added new text in the Discussion on page 17: “The limitation of our study is that we did not determine what was the percentage of histone H4 molecules that were modified by PAD4 during citrullination in vitro and which were the critical arginines implicated. While likely not all molecules were citrullinated, the impact of the modifications on reducing histone H4 toxicity was significant. Similarly, it will be important to determine the fraction of histones that are citrullinated during NETosis and whether this fraction and the identity of histones modified vary with stimulants inducing NET formation.”

4. Please provide the product number and any lot numbers of the antibodies purchased for your study.

We have now added the catalog number and lot number information for antibodies used in our study on pages 7 and 8.

5. At this time, we request that you please report additional details in your Methods section regarding animal care, as per our editorial guidelines:

(a) Please state the number of mice used in the study

We added the number of wild type and PAD4 knockout mice used in our study in the Methods on page 4.

(b) Please provide details of animal welfare (e.g., shelter, food, water, environmental enrichment)

We added the details of animal welfare/housing in our Method section on page 4.

(c) Please include the method of euthanasia; please specify whether mice were recovered following retro-orbital bleeding.

Mice are deeply anesthetized to effect with isoflurane, and they are terminally bled via the retro orbital sinus before being sacrificed by cervical dislocation. We specified this in our Method section on page 5.

6. In your Methods section, please provide additional details regarding the cell lines used in your study and ensure you have described the source.

HL-60 cells were purchased from ATCC. We provided the source and its catalog number in our revision on page 4.

7. Please provide additional details regarding participant consent.

In the ethics statement in the Methods and online submission information, please ensure that you have specified what type you obtained (for instance, written or verbal, and if verbal, how it was documented and witnessed).

If your study included minors, state whether you obtained consent from parents or guardians.

If the need for consent was waived by the ethics committee, please include this information.

We specified that written consent was acquired in our Method section on page 4. We do not have any minors in our study.

8. PLOS ONE now requires that authors provide the original uncropped and unadjusted images underlying all blot or gel results reported in a submission’s figures or Supporting Information files. This policy and the journal’s other requirements for blot/gel reporting and figure preparation are described in detail at https://journals.plos.org/plosone/s/figures#loc-blot-and-gel-reporting-requirements and https://journals.plos.org/plosone/s/figures#loc-preparing-figures-from-image-files. When you submit your revised manuscript, please ensure that your figures adhere fully to these guidelines and provide the original underlying images for all blot or gel data reported in your submission. See the following link for instructions on providing the original image data: https://journals.plos.org/plosone/s/figures#loc-original-images-for-blots-and-gels.

We have provided the original uncropped and unadjusted images for all blot results reported in our manuscript and supplementary materials. It is named as S4_raw_images.

9. PLOS requires an ORCID iD for the corresponding author in Editorial Manager on papers submitted after December 6th, 2016. Please ensure that you have an ORCID iD and that it is validated in Editorial Manager. To do this, go to ‘Update my Information’ (in the upper left-hand corner of the main menu), and click on the Fetch/Validate link next to the ORCID field. This will take you to the ORCID site and allow you to create a new iD or authenticate a pre-existing iD in Editorial Manager. Please see the following video for instructions on linking an ORCID iD to your Editorial Manager account: https://www.youtube.com/watch?v=_xcclfuvtxQ.

We linked the ORCID ID of the corresponding author to the submission system.

10. Please include captions for your Supporting Information files at the end of your manuscript, and update any in-text citations to match accordingly.

We have now moved Supporting Information and its captions to the end of the manuscript, after the Conclusion and before the Acknowledgment, on pages 18 and 19.

Review Comments to the Author:

Reviewer #1: The paper analyzes the ability of “native” vs citrullinated histones to induce NETosis in mouse and human neutrophils and in HL60 cells. The results indicate that histone H4 induces NETosis mediating calcium influx and PADi4 activation, while citrullinated H4 has a lower activity, suggesting that citrullinated histones in NET may act as negative feedback mechanism in NET production.

On the whole, the data presented in the paper are of great interest and relevance; only a minor point can be raised.

We thank the reviewer for his/her positive comments on the manuscript.

Under a quantitative point of view, is NET induction by native histones comparable to what obtained with other “physiological” stimuli (for example lps)? 

Thank you for bringing up this interesting question. The NET induction by native histones is comparable to other physiological stimuli. For example, LPS induced NET formation is around 9.5% while histone H4 induced NET formation is around 12%. We also added this comparison to our revised manuscript for information. It is in lines 219~220.

Reviewer #2: The manuscript submitted by Shi et al. addresses the role of H4 citrullination in NET formation. The topic is interesting and the results suggest that the release of citrullinated H4 may constitute a physiological mechanism to slow down the process of NET formation. The manuscript is well written, but it would need additional controls and changes before considering its publication in PlosOne.

Specific comments:

1. In line 192 the authors state that citrullinated H1 is released into the extracellular space along native H4. The authors would need to demonstrate this statement showing an experiment where native H4 is shown to be released into the extracellular space. Moreover, it would be really helpful to determine the proportion H4 modified by citrullination. These results should be added to Figure 1, or as an additional figure.

We thank the reviewer for bringing up this point. That native histone H4 is associated with released NETs is a known fact; therefore, we provided two citations (Ref 1 and 30 in line 198) to illustrate this. It would have been technically difficult for us to determine the percentage of histone H4 molecules citrullinated after PAD4 reaction. We now provide this as a limitation to our studies on page 17 and suggest other points in this direction that could be done in the future: “The limitation of our study is that we did not determine what was the percentage of histone H4 molecules that were modified by PAD4 during citrullination in vitro and which were the critical arginines implicated. While likely not all molecules were citrullinated, the impact of the modifications on reducing histone H4 toxicity was significant. Similarly, it will be important to determine the fraction of histones that are citrullinated during NETosis and whether this fraction and the identity of histones modified vary with stimulants inducing NET formation.”

2. The statistical analysis shown in the figures must be revised to include all possible pairs. For instance, Figure 2B does not show if the percentage of NET formation in the presence of Cla is significant when compared to the control cells. The same problem can be observed in all the figures in panels with more than two bars.

We have now added the statistical analysis to all the comparisons in our figures in the revised manuscript.

3. The extent of in vitro citrullination must be determined to ensure that all H4 has been modified in the reaction conditions. These results must be included in Figure S2.

Please see point 1.

4. Regarding the manuscript format, higher resolution images are needed, especially for immunofluorescence and the supplementary file must be revised because at the moment previous changes are marked in red within the document.

For mouse NET immunofluorescence, we replaced Fig 2E and G with higher contrast images that improved the H3Cit staining resolution. Our apologies, we must have forgotten to turn off the tracked changes in our initial submission. We will make sure this does not occur again! We thank the reviewer for carefully scrutinizing our manuscript, and we believe the review process improved its quality.

---

## [Decision Letter · Decision Letter 1]

3 May 2021

Neutrophil stimulation with citrullinated histone H4 slows down calcium influx and reduces NET formation compared with native histone H4

PONE-D-21-02995R1

Dear Dr. Wagner,

We’re pleased to inform you that your manuscript has been judged scientifically suitable for publication and will be formally accepted for publication once it meets all outstanding technical requirements.

Kind regards,

Michel Simon, Ph. D.

Academic Editor

PLOS ONE

Additional Editor Comments (optional):

Reviewers' comments:

Reviewer's Responses to Questions

**Comments to the Author**

1. If the authors have adequately addressed your comments raised in a previous round of review and you feel that this manuscript is now acceptable for publication, you may indicate that here to bypass the “Comments to the Author” section, enter your conflict of interest statement in the “Confidential to Editor” section, and submit your "Accept" recommendation.

Reviewer #1: All comments have been addressed

Reviewer #2: All comments have been addressed

2. Is the manuscript technically sound, and do the data support the conclusions?

Reviewer #1: Yes

Reviewer #2: Yes

3. Has the statistical analysis been performed appropriately and rigorously? 

Reviewer #1: Yes

Reviewer #2: Yes

4. Have the authors made all data underlying the findings in their manuscript fully available?

Reviewer #1: Yes

Reviewer #2: Yes

5. Is the manuscript presented in an intelligible fashion and written in standard English?

Reviewer #1: Yes

Reviewer #2: Yes

6. Review Comments to the Author

Reviewer #1: The point raised has been addressed. No further modifications are required and the paper can be accepted in the present form.

Reviewer #2: Shi et al. have addressed all the comments, although they didn’t determine the extent of H4 citrullination as requested. However, they provided an understandable reason for this issue and highlighted this limitation within the discussion. As a whole, the manuscript is significantly improved in respect of the original version and is now suitable for publication in PLosOne.

7. PLOS authors have the option to publish the peer review history of their article (what does this mean?). If published, this will include your full peer review and any attached files.

Reviewer #1: **Yes: **Paola Migliorini

Reviewer #2: No

---

## [Editor Report · Acceptance letter]

7 May 2021

PONE-D-21-02995R1 

Neutrophil stimulation with citrullinated histone H4 slows down calcium influx and reduces NET formation compared with native histone H4 

Dear Dr. Wagner:

I'm pleased to inform you that your manuscript has been deemed suitable for publication in PLOS ONE. Congratulations! Your manuscript is now with our production department. 

Kind regards, 

on behalf of

Dr. Michel Simon 

Academic Editor

PLOS ONE